

# Genome-wide sequence identification and expression analysis of $N^6$-methyladenosine demethylase in sugar beet (*Beta vulgaris* L.) under salt stress

Jie Cui[1,*],  Junli Liu[1,*],  Junliang Li[1,2],  Dayou Cheng[1] and  Cuihong Dai[1]

[1] Harbin Institute of Technology, Harbin, Heilongjiang, China
[2] College of Life and Environmental Science, Wenzhou University, Wenzhou, Zhejiang, China
[*] These authors contributed equally to this work.

## ABSTRACT

In eukaryotes, $N^6$-methyladenosine (m⁶A) is the most abundant and highly conserved RNA modification. *In vivo*, m⁶A demethylase dynamically regulates the m⁶A level by removing the m⁶A marker where it plays an important role in plant growth, development and response to abiotic stress. The confirmed m⁶A demethylases in *Arabidopsis thaliana* include ALKBH9B and ALKBH10B, both belonging to the ALKB family. In this study, BvALKB family members were identified in sugar beet genome-wide database, and their conserved domains, gene structures, chromosomal locations, phylogeny, conserved motifs and expression of *BvALKB* genes were analyzed. Almost all BvALKB proteins contained the conserved domain of 2OG-Fe II-Oxy. Phylogenetic analysis suggested that the ten proteins were clustered into five groups, each of which had similar motifs and gene structures. Three *Arabidopsis* m⁶A demethylase-homologous proteins (BvALKBH6B, BvALKBH8B and BvALKBH10B) were of particular interest in our study. Expression profile analysis showed that almost all genes were up-regulated or down-regulated to varying degrees under salt stress. More specifically, *BvALKBH10B* homologous to *AtALKBH10B* was significantly up-regulated, suggesting that the transcriptional activity of this gene is responsive to salt stress. This study provides a theoretical basis for further screening of m⁶A demethylase in sugar beet, and also lays a foundation for studying the role of ALKB family proteins in growth, development and response to salinity stress.

## INTRODUCTION

The nucleotide $N^6$-methyladenosine (m⁶A) is the most abundant modification in mRNA among all higher eukaryotes, manifested as methylation at the sixth nitrogen ($N$) of adenosine , which has been a specific focus for epigenetic studies in recent years (*Huang & Yin, 2018*; *Huang et al., 2021*; *Scarrow, Chen & Sun, 2020*; *Zhou et al., 2020*). Previous studies have shown that m⁶A in addition to methyltransferase complex (METTL3, METTL14, WTAP, *etc.*), demethylases (FTO, ALKBH5, *etc.*) and RNA binding proteins (YTHDF1/2/3, YTHDC1/2, *etc.*) (*Desrosiers, Friderici & Rottman, 1974*; *Ortega et al.,*

Corresponding author
Jie Cui, cuijie@hit.edu.cn

*2003*; *Jia et al., 2011*), constitute a reversible and dynamic co-regulation process (*Miao et al., 2020*). In animals, genes encoding m⁶A-related proteins have been identified and characterized (*Wei, Gershowitz & Moss, 1976*; *Levis & Penman, 1978*), and their important role in animal development has been demonstrated, but the function of these proteins in plants is yet to be fully elucidated. Generally, m⁶A is enriched near the stop codon and the 3′-untranslated region, occurs within long introns and at transcription start sites (*Meyer et al., 2012*), which are common in mammals. In *Arabidopsis thaliana*, m⁶A is enriched near the start codon of many genes suggesting that this epigenetic modification may play a role in a plant-specific context (*Luo et al., 2013*; *Wan et al., 2015*). Recently, numerous molecular studies focusing on m⁶A methylation have demonstrated its role in eukaryotic transcriptome regulation, RNA stability, and translation efficiency (*Niu et al., 2013*; *Pan, 2013*; *Yue et al., 2019*). In plants, many proteins are involved in regulating the formation of cells and tissues (*Zhong et al., 2008*; *Shen et al., 2016*; *Bhat et al., 2020*; *Scutenaire et al., 2018*), while others regulate the expression of drought and high temperature signal-related genes (*Zhao et al., 2014*; *Lu et al., 2020*), which play a significant role in the response to various stresses.

The reversibility of RNA methylation has been demonstrated to be achieved by demethylases (*Jia et al., 2011*). Proteins identified as m⁶A demethylases belong to the ALKB family and contain highly conserved synthase-like domains. The m⁶A demethylases found in mammals mainly include obesity-related genes (FTO) and ALKBH5 (*Jia, Fu & He, 2013*; *Liu & Jia, 2014*). The unique C-terminal long loop structure of FTO genes may indicate its function of promoting protein-protein or protein-RNA interactions. The demethylase ALKBH5 can modify m⁶A to produce adenosine (A) directly, without intermediates such as FTO (*Mauer et al., 2017*; *Wei et al., 2018*). Due to differences in tissue and substrate specificity, FTO and ALKBH5 play different roles in mRNA processing and metabolism. Studies have shown that FTO can regulate splicing and maturation of precursor RNA *via* binding to SRSF2, while ALKBH5 is associated with the nuclear transport of mRNA (*Zhao et al., 2014*).

Bioinformatic analysis revealed that there are 14 ALKB homologous proteins in *Arabidopsis*, among which ALKBH9A, ALKBH9B, ALKBH9C, ALKBH10A and ALKBH10B had the most similar amino acid sequences to ALKBH5. Proteins that have been confirmed as m⁶A demethylases include ALKBH9B and ALKBH10B the latter is highly abundant in all tissues, but accumulates to the highest abundance in floral tissues. It has a specific catalytic function on m⁶A-modified mRNA, and experiments have shown that ALKBH10B can mediate the early flowering transition by regulating the demethylation of *FT*, *SPL3* and *SPL9* (*Duan et al., 2017*). As the only ALKBH5-homologous protein in the cytoplasm, ALKBH9B was found to be responsible for removing $N^6$-methyladenosine from ssRNA *in vitro* and participating in mRNA silencing or degradation. In addition, ALKBH9B also plays a role in plant protection against specific viral pathogens where it can impair the systemic invasion by increasing the relative abundance of m⁶A in the *Alfalfa mosaic virus* (AMV) genome (*Martínez-Pérez et al., 2017*). However, studies on m⁶A demethylase in other plant species are lacking.

Previous studies have demonstrated the role of some ALKBH members in plant growth and development. The function of demethylase in the response to stress has been extensively studied in *Arabidopsis*. ALKBH9A was found to be highly expressed in roots under salt stress, and ALKBH10A was significantly down-regulated under heat stress (*Růžič et al., 2017*). Under drought, cold or ABA treatment, ALKBH1 levels were significantly up-regulated, while ALKBH6, ALKBH8B and ALKBH10A expression was down-regulated (*Hu, Manduzio & Kang, 2019*), indicating that ALKBH members may play an important role in the abiotic stress response. In recent studies, it was found that ALKBH6 could bind to m$^6$A marked mRNA and remove the mark in *Arabidopsis*, and therefore, ALKBH6 may be a potential m$^6$A demethylase. Under drought or heat but not salt stress, the survival rate of the *alkbh6* mutant was lower than that of the wild type. In addition, ALKBH6 affected the ABA response by regulating the expression of genes related to ABA signaling (*Huong, Ngoc & Kang, 2020*). These results suggest that RNA demethylation plays a crucial role in plant responses to abiotic stress.

Sugar beet is one of the most important sugar-producing crops, and its yield and quality are of great significance in agricultural production. In China, the saline-alkali soil is often encountered in the sugar beet production areas (*Liu & Wang, 2021*; *Yang et al., 2012*). Although sugar beet has a degree of salt tolerance, excessive salt will affect its germination, growth and yield. Therefore, the analysis of sugar beet m$^6$A will be helpful to understand its transcriptional modification and expression regulation, and reveal its salt-tolerant mechanism to breed new stress resistant strains. Although m$^6$A demethylase has been proved to be involved in the response to abiotic stress (*Hu et al., 2021*), so far there has been no specific analysis of the function of this enzyme in sugar beet under salt stress. In this study, bioinformatic analysis of m$^6$A demethylase was carried out based on the sugar beet genome database, and demethylase genes related to salt treatment were identified. The data provide new avenues for breeding salt tolerant sugar beet varieties.

## MATERIALS & METHODS

### Materials

The salt-tolerant strain "O"68 of sugar beet was used as the experimental material in this experiment (*Shi et al., 2008*). The seeds were soaked in running water for 12 h, disinfected and sown onto wet sponge and cultured in the dark for 2 days. After germination, the seedlings were transferred to culture pots (43.5 × 20 × 14 cm, 10 plants per pot) containing quarter-strength Hoagland solution. The seedlings were cultivated under 16/8 light photoperiod at 24 °C/18 °C day/night temperature in a phytotron (Friocell 707, Germany). After the formation of three pairs of true leaves, 300 mM NaCl was used to replace the nutrient solution for 24 h, and other conditions were kept unchanged. The control group was treated with nutrient solution without salt treatment. After the salt stress treatment, leaves and roots were sampled. Plant samples from the same treatment were premixed and divided into 0.2 g small packages, immediately frozen in liquid nitrogen and stored at −80 °C until used for analysis.

## Screening and identification of sugar beet m$^6$A demethylases

The whole genome database of sugar beet has been published (http://bvseq.molgen.mpg.de/index.shtml). The encoding motif sequence of the demethylase conserved domain 2OG-Fe II-oxy (PF13532) was downloaded from the Pfam database. The $e$-value $<1e^{-5}$ was set on HMMER (http://www.hmmer.org/) and the beet genome-wide database was searched. The Pfam online tool was used to analyze the domains of candidate proteins, and the proteins with the conserved domain were considered to be BvALKB proteins. Multiple sequence alignment of BvALKB proteins was performed using DNAMAN7.0 and its conserved domain was identified using Weblogo (http://weblogo.berkeley.edu/).

## Bioinformatic analysis of BvALKB family

ExPASY (https://web.expasy.org/protparam/) was used to analyze the physical and chemical properties of proteins, including the average molecular weight, isoelectric point and average number of amino acids (*Gasteiger et al., 2003*). Protein subcellular localization was predicted by CELLO (http://cello.life.nctu.edu.tw/). MapGene2Chrom (http://mg2c.iask.in/mg2c_v2.0/) was used to map the position of each gene on a chromosome. MEME (http://meme-suite.org/tools/meme) was used to predict protein motifs (*Bailey et al., 2006*), and the number of motifs was set to 20, with other parameters for tacit recognition. Gene intron and exon structures were analyzed in Splign (https://www.ncbi.nlm.nih.gov/sutils/splign/splign.cgi?textpage=online{&}level=form). A phylogenetic tree (1000 replicates) was constructed by neighbor-joining method using MEGA7 for protein sequence progression and multi-sequence alignment between *Arabidopsis* and sugar beet (*Kumar, Stecher & Tamura, 2016*).

## Expression analysis of BvALKB genes and gene cloning

All samples were ground in liquid nitrogen. Total RNA was extracted using TRIzol reagent and the concentration of RNA was determined using a MicroDrop spectrophotometer. Total RNA was reverse-transcribed into cDNA by using PrimeScript$^{TM}$ II first strand cDNA Synthesis Kit (TaKaRa, Japan). In order to detect the gene expression level, qRT-PCR was performed using the CFX96 real-time system and the iTaq$^{TM}$ Universal SYBR Green Supermix Kit (BIO-RAD, USA). The primers were designed using Primer 5 and their sequences are listed in Table 1. Referring to our previous studies (*Li et al., 2020*), *UBQ5* and *PP2A* were used as internal controls for leaves, and *25S rRNA* and *PP2A* were used as internal controls for roots. All experiments were repeated at least three times. Data analysis was performed by $2^{-\triangle\triangle Ct}$ method. The relative expression of each gene was presented as mean $\pm$ standard deviation. The PCR primers were designed for *BvALKBH10B* cloning as follows:

F: 5′-GGAATTCATGTCGCCGGCGGCGGGACCATTGT-3′,
R: 5′-GGGATCCTCACATTATCCTTCCTTCCACACCTGGGTCAGACATGGT-3′.

## Data treatment and statistical analysis

For the data of qPCR analysis, the mean and SD were calculated from three repeats of each treatment, and the differences were analyzed by SPSS 20.0 using independent-samples $t$-test ($p < 0.05$).

**Table 1  Primer sequences of BvALKB genes.**

| Gene | Forward primer (5′-3′) | Reverse primer (5′-3′) |
|------|------------------------|------------------------|
| UBQ5 | TCTGCTGGAAGAGCCTTTGG | TTGTCGCCGCTCTTTACACT |
| 25S rRNA | AGACAAGAAGGGGCAACGAG | CACATTGGACGGGGCTTTTC |
| BvPP2A | TCGTGTCCAAGAAGTGCCTC | CACAACGGTCATCAGGGTCA |
| BvALKBH1B | AGGGAATGCTTTCATGGGGT | CTCGAACCAAGCTATCCGGG |
| BvALKBH2B | GTACTTCCAATAAAACGTCACCGT | GTTTTCAGATGAATCACATGTGCCA |
| BvALKBH3B | TAGCTCGGAACAGGCGAAAA | TGTGGAATTGCCGGTGGTAT |
| BvALKBH4B | CATATTCTCCAGGCGGTCCA | GGCGTTCACAACCAAAGGAA |
| BvALKBH5B | AGTCCGGAGGAGTCCAGAAA | AGGTCCTGTTCTGACCTTGC |
| BvALKBH6B | AAACGGCAGCTTATGGAACG | ATGGGAGGCAAGGGATCAAC |
| BvALKBH7B | GGCTTTACAGTCGGCTCTGT | GTCAGCCAAGGAGGCAAGTC |
| BvALKBH8B | TTCCCTTGCCTGTTGGATCG | GCAAAATACACAGGCCGCTT |
| BvALKBH9B | TACCAGCCAGGTGAGGGTAT | CGAGCATCGCCTGACATGAT |
| BvALKBH10B | GGTGGGAAACAAGGGAGGAG | CCTCATGTGAGCCTGTGTCA |

# RESULTS

## Identification of sugar beet m$^6$A demethylases

The seed sequence of the conserved domain (PF13532) was downloaded from Pfam and used as search bait in the beet genome database by HMMER. A total of 10 homologous proteins were identified, and they were named BvALKBH1B through 10B (Table 2). The e-value of all other proteins was less than 1e$^{-5}$ except BvALKBH10B, where it was 0.016. The ten proteins were blastp-aligned with the NCBI database, as shown in Table 2. Information from NCBI suggests that BvALKBH2B, BvALKBH3B, BvALKBH8B and BvALKBH10B have not been described before and belonged to new ALKB family members while the other proteins have been confirmed to belong to the ALKB family.

The domains of the ten candidate proteins were analyzed by Pfam (Fig. 1). All proteins have 2OG-Fe II-Oxy domain except for BvALKBH10B, indicating that these domains are highly conserved. In terms of domain distribution, the domains of BvALKBH7B were at the N-terminus, and the domains of BvALKBH1B, BvALKBH2B, BvALKBH3B, BvALKBH4B, BvALKBH5B, and BvALKBH9B were all at the C-termini. The RRM domain of BvALKBH5B was related to mRNA and rRNA processing, RNA output and RNA stability by query. However, due to low sequence similarity, the e-value of BvALKBH10B in Pfam database comparison was 0.023, and it has a high possibility of possessing the 2OG-Fe II-Oxy domain, so it is regarded as a member of this family for subsequent analysis.

The alignment results of DNAMAN7.0 showed certain homology but low conservation in the domain sequences (Fig. 1). The homology was very high at the sites 162, 212, 215, 222, 255, 259, etc., which might be related to the function of the domain and amino acids at these specific locations.

## Analysis of physicochemical properties of BvALKB proteins

Physical and chemical properties analysis showed that the average length of the coding region of the ten genes was 1,260 bp (and for all numbers of 1,000), the average number of

**Table 2  Basic information of BvALKB.**

| Gene name | BvALKB name | NCBI reference sequence | Gene ID | Description |
|---|---|---|---|---|
| *Bv6_150770_huzh* | BvALKBH1B | XM_010684461.2 | 104897561 | PREDICTED: Beta vulgaris subsp. vulgaris alpha-ketoglutarate-dependent dioxygenase alkB (LOC104897561) |
| *Bv7_169620_pkhc* | BvALKBH2B | XM_010686965.2 | 104899719 | PREDICTED: Beta vulgaris subsp. vulgaris hypothetical protein |
| *Bv7_157650_ryeg* | BvALKBH3B | XM_010685256.2 | 104898211 | PREDICTED: Beta vulgaris subsp. vulgaris uncharacterized LOC104898211 |
| *Bv8_184320_kacr* | BvALKBH4B | XM_010688312.2 | 104900793 | PREDICTED: Beta vulgaris subsp. vulgaris DNA oxidative demethylase ALKBH2 |
| *Bv5_102160_pgse* | BvALKBH5B | XM_010678383.2 | 104892444 | PREDICTED: Beta vulgaris subsp. vulgaris alkylated DNA repair protein alkB homolog 8 |
| *Bv3_051230_eskg* | BvALKBH6B | XM_010673069.2 | 104888178 | PREDICTED: Beta vulgaris subsp. vulgaris RNA dementhylase ALKBH5 |
| *Bv4_083160_sqec* | BvALKBH7B | XM_010676670.2 | 104891030 | PREDICTED: Beta vulgaris subsp. vulgaris alpha-ketoglutarate-dependent dioxygenase alkB homolog 6 |
| *Bv6_130050_njrf* | BvALKBH8B | XM_010681565.2 | 104895138 | PREDICTED: Beta vulgaris subsp. vulgaris uncharacterized LOC104895138 |
| *Bv7_164580_swwm* | BvALKBH9B | XM_010686203.2 | 104899068 | PREDICTED: Beta vulgaris subsp. vulgaris alkylated DNA repair protein alkB homolog 8 |
| *Bv7_179400_uxaj* | BvALKBH10B | XM_010698038.2 | 104908870 | PREDICTED: Beta vulgaris subsp. vulgaris hypothetical protein |

amino acids per protein was 416 (260–584), the average molecular weight was 46.41 kDa (28.91–64.97 kDa), and the average isoelectric point was 7.12 (5.11–9.02) (Table 3).

## Chromosomal localization of genes

Sugar beet is diploid with 2n =18 chromosomes. Chromosome localization analysis showed that BvALKBH6B was located on chromosome 3, BvALKBH7B was located on chromosome 4, BvALKBH5B was located on chromosome 5, BvALKBH1B and BvALKBH8B were located on chromosome 6, BvALKBH2B, BvALKBH3B and BvALKBH9B were located on chromosome 7, and BvALKBH4B was located on chromosome 8 (Fig. 1). Analysis of BvALKBH10B showed that it was located on chromosome 7, but lacked localization information, which may be due to gaps in genome assembly. These results indicate that the chromosome distribution of BvALKB genes is relatively scattered, without cluster distribution and lack of chromosome preference.

## Phylogenetic relationships and gene structure analysis of BvALKB

Multiple sequence alignment was performed on 14 ALKB family proteins of *Arabidopsis* and ten proteins of sugar beet using MEGA7, and the alignment diagram of protein local domain was analyzed (Fig. 2). The results showed that the differences in amino acid sequences of ALKB proteins led to low domain homology and therefore belonged to different subclasses, but remained highly conserved at individual sites.

Then a phylogenetic tree (1000 replicates) was constructed using neighbor-joining method to observe the evolutionary relationships between *Arabidopsis* and sugar beet

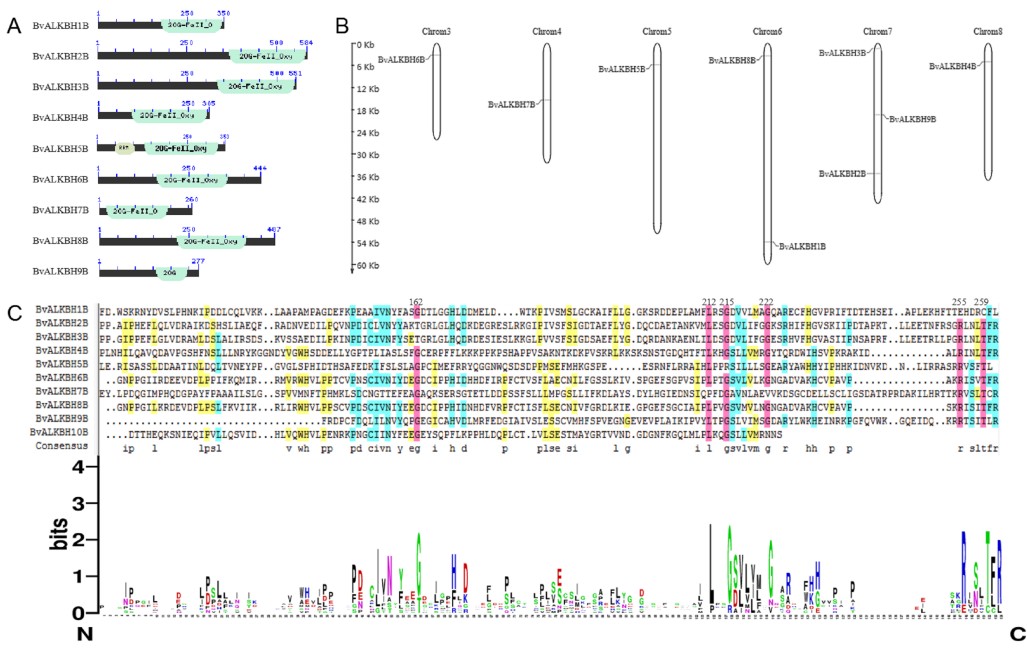

**Figure 1** Conserved domain analysis and chromosome localization of BvALKBs. (A) Conserved domain analysis of BvALKB proteins. (B) Chromosome localization of BvALKB genes. (C) Sequence analysis of the conserved domain in BvALKB proteins.

**Table 3** Physical and chemical properties analysis of BvALKB proteins.

| BvALKB name | ORF(bp) | Amino acid | Molecular weight(Da) | PI |
| --- | --- | --- | --- | --- |
| BvALKBH1B | 1053 | 350 | 39477.03 | 7.13 |
| BvALKBH2B | 1755 | 584 | 64923.52 | 7.15 |
| BvALKBH3B | 1656 | 551 | 60969.22 | 8.74 |
| BvALKBH4B | 1018 | 305 | 34594.96 | 9.02 |
| BvALKBH5B | 1062 | 353 | 39620.72 | 6.53 |
| BvALKBH6B | 1335 | 444 | 49776.81 | 8.86 |
| BvALKBH7B | 783 | 260 | 28912.06 | 5.70 |
| BvALKBH8B | 1464 | 487 | 54949.39 | 6.62 |
| BvALKBH9B | 834 | 277 | 30792.26 | 5.11 |
| BvALKBH10B | 1641 | 546 | 60084.61 | 6.30 |

proteins (Fig. 3). Most of the bootstrap values were greater than 70, indicating high reliability. The proteins were divided into five categories: Class I (AtALKBH9-like) included BvALKBH6B and BvALKBH8B, which are similar to AtALKBH9; Class II (AtALKBH10-like) only contained BvALKBH10B, which is similar to AtALKBH10; Class III (AtALKBH2-like) contained only one BvALKB protein; Class IV (AtALKBH6/8-like) consisted of BvALKBH5B, BvALKBH7B and BvALKBH9B; Three members were assigned to Class V (AtALKBH1-like), including BvALKBH1B, BvALKBH2B and BvALKBH3B (Fig. 3). AtALKBH9B and AtALKBH10B in the first two classes were confirmed to be m[6]A

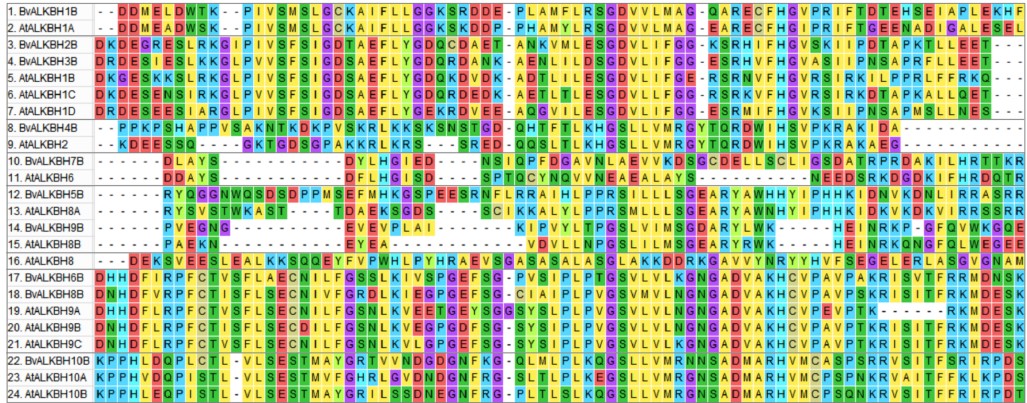

**Figure 2** **Multiple sequence alignment between BvALKB and AtALKB proteins.** Different colors represent residues with different characteristics.

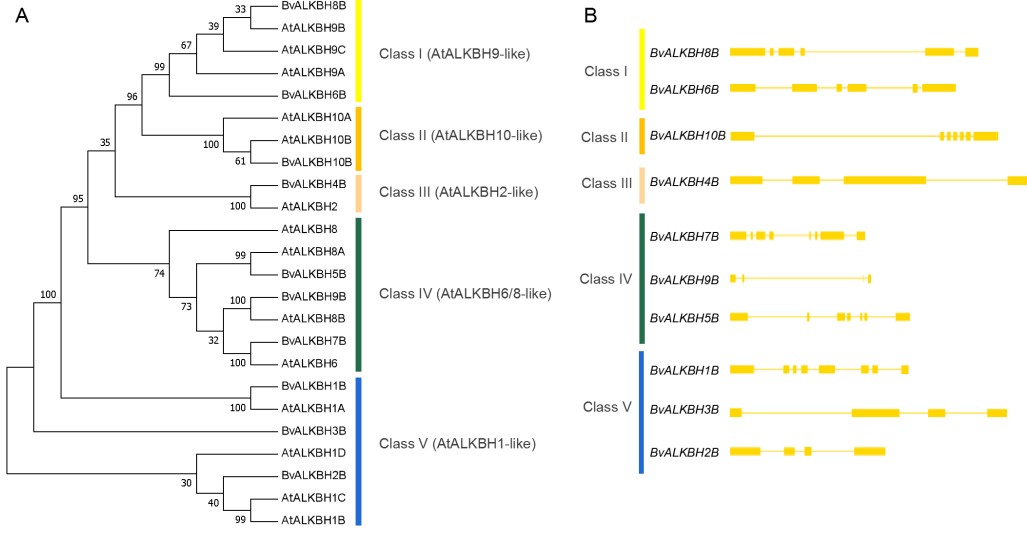

**Figure 3** **Phylogenetic relationships and gene structures of BvALKBs.** (A) Phylogenetic relationships of BvALKB and AtALKB proteins. The gene class is represented in a different color on the right side of the rootless tree. (B) Gene structures of BvALKB genes. Exon/intron structures of the BvALKB genes are represented in differnent ways. Exons and introns are represented by yellow box and lines, respectively.

demethylases, so BvALKBH6B, BvALKBH8B and BvALKBH10B are also likely to have demethylase function, however, future experimental validation is required to confirm the function of these three ALKBH proteins.

The gene structure analysis revealed that genes within the same group showed similar intron and exon organization. The *BvALKBH6B* and *BvALKBH8B* of Class I had six exons, whereas *BvALKBH2B* and *BvALKBH3B* of Class V had four exons, based on their sequence similarity. The *BvALKBH5B* and *BvALKBH7B* in Class IV were similar in structure, although

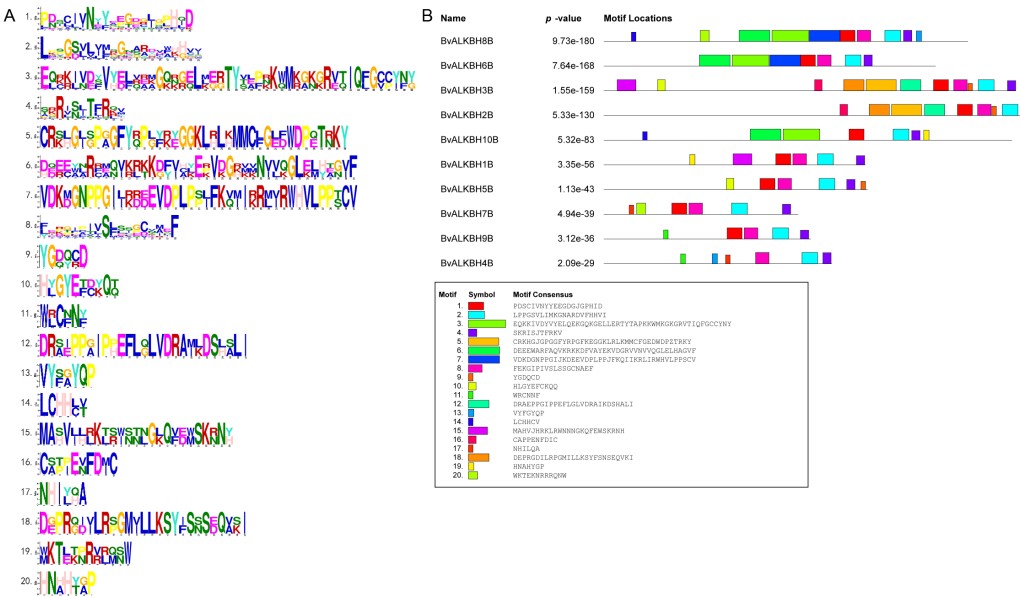

**Figure 4 Motif Analysis of BvALKB proteins.** (A) Motifs in BvALKB proteins. The motifs were arranged according to the e-value from small to large, the letters in each motif were amino abbreviation. The size of the letter represented the saliency of the amino acid in the motif. The larger the letter, the higher the saliency, which is, the higher the frequency at which the amino acid appears in the same position in the same motif in different sequences. (B) Analysis of BvALKB proteins motifs. The different color blocks correspond to different motifs. The width of the color block is the length of the motif. The height of the color block represents the saliency of the motifs in the sequence. The higher the saliency, the more able to match the predicted motifs.

the number of exons was different. The structures of *BvALKBH1B* and *BvALKBH9B* were different to varying degrees from those of the above genes (Fig. 3).

## Motif analysis and subcellular localization prediction of BvALKB proteins

The motif analysis of BvALKB proteins is shown in Fig. 4. In general, the ten proteins (except BvALKBH10B) had the motifs 1, 2, 4, and 8, which are probably important components of the 2OG-Fe II-Oxy domain. Proteins belonging to the same group had similar motif composition. BvALKBH6B, BvALKBH8B and BvALKBH10B homologous to AtALKBH9B/10B differed from other proteins in motif composition because they had closely connected motif 3 and motif 6, which may be related to the demethylase function.

The scores of different locations of CELLO predicted proteins showed that most of the proteins were located in the nucleus and mainly performed function of demethylation in the nucleus (Table 3). The proteins BvALKBH10B and BvALKBH7B were both located to the cytoplasm. However, BvALKBH7B was additionally predicted to be in the extracellular region presumably performing other extra-nuclear functions.

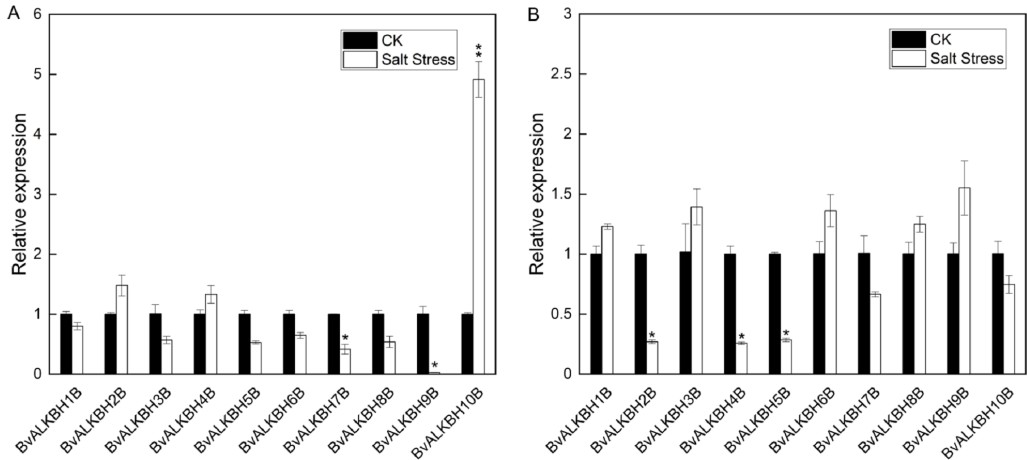

**Figure 5** Expression analysis of BvALKB genes under salt stress. (A) Expression analysis of BvALKB genes in leaves in response to salinity stress. (B) Expression analysis of BvALKB genes in roots in response to salinity stress. Error bars indicate standard deviation. Asterisks (* and **) indicate statistically significant differences, as determined by Student's $t$ tests, at $p < 0.05$ and $p < 0.01$, respectively.

## Quantitative analysis of BvALKB genes in sugar beet under salt stress

$N^6$-methyladenosine plays an important role in response to abiotic stresses. In order to understand the changes in the potential $m^6A$ demethylase genes in sugar beet under salt stress, we compared the expression levels of the genes under normal and salt stress conditions. The phenotypic changes of sugar beet grown to the stage of three pairs of true leaves were observed under 300 mM salt stress, and the expression of each gene was analyzed by qRT-PCR.

In leaves, all genes were up-regulated or down-regulated to varying degrees except *BvALKBH1B*. *BvALKBH2B*, *BvALKBH4B* and *BvALKBH10B* were up-regulated and in particular *BvALKBH10B* was highly up-regulated (Fig. 5). *BvALKBH3B*, *BvALKBH5B*, *BvALKBH6B*, *BvALKBH7B*, *BvALKBH8B* and *BvALKBH9B* were down-regulated, and *BvALKBH9B* showed a significant and striking degree of expression down-regulation. In roots, *BvALKBH1B*, *BvALKBH3B*, *BvALKBH6B*, *BvALKBH8B* and *BvALKBH9B* were up-regulated, while the other five genes were down-regulated. *BvALKBH2B*, *BvALKBH4B* and *BvALKBH5B* were significantly down-regulated (Fig. 5). Under salt stress, the expression levels of BvALKBH7B, BvALKBH9B and BvALKBH10B in leaves were significantly affected, whereas that of BvALKBH2B, BvALKBH4B and BvALKBH5B in roots was only mildly modified by stress treatment. These results indicate that these genes had tissue-specific activity in regulating the response to salt stress in sugar beet.

Since BvALKBH10B and AtALKBH10B were highly homologous and the expression level of BvALKBH10B changed significantly under salt stress, we cloned and sequenced the BvALKBH10B gene from Beet "O" 68. The sequencing results were submitted to the Genbank database (MZ358117), which was consistent with the whole genome database of sugar beet.

## DISCUSSION

Soil salinization has become a global problem. In China, saline-alkali land is mainly distributed in northern China, and highly coincides with the sugar beet production area, which puts forward higher requirements for sugar beet salt tolerance. Previous studies have shown that members of the ALKB protein family are involved in plant growth and development and abiotic stress responses, especially the proteins confirmed as $m^6A$ demethylases. However, the ALKB family members in sugar beet have not been studied. Therefore, bioinformatic and quantitative methods were used to study the response of sugar beet ALKB genes to salt stress, which provided a theoretical basis for the screening of demethylase in sugar beet.

Through the beet genome-wide analysis, we found ten BvALKB family proteins. The number was similar to *Arabidopsis* (14) and rice (12), but much less than that of wheat (29) and quinoa (27) (*Yue et al., 2019*), which might have resulted from different copy numbers during plant evolution (*Miao et al., 2021*).

Phylogenetic analysis can efficiently identify the homology and evolutionary relationships of proteins. The phylogenetic tree of BvALKB proteins and AtALKB proteins was constructed using neighbor-joining method. All proteins were divided into five classes. Class I contained AtALKBH9A/9B/9C proteins, and two BvALKB proteins (BvALKBH6B and BvALKBH8B) belonged to this group. Class II contained AtALKBH10A/10B proteins, with only BvALKBH10B belonging to it. Proteins were considered orthologous to *Arabidopsis* AtALKBH9 and AtALKBH10 if they possessed a function similar to $m^6A$ demethylation (*Yue et al., 2019*). Therefore, BvALKBH6B, BvALKBH8B and BvALKBH10B were putatively considered as potential $m^6A$ demethylases. A previous study demonstrated that ALKBH2 is an important enzyme for protecting *Arabidopsis* against damaging DNA methylation changes, and suggested that its homologues in other plants may have a similar function (*Meza et al., 2012*). Thus, BvALKBH4B belonging to Class III (AtALKBH2-like) may be involved in protecting plants from DNA methylation damage. Mammalian and *Arabidopsis* ALKBH8 were demonstrated as tRNA hydroxylases targeting 5-methoxycarbonylmethyl-modified uridine (mcm5U) at the wobble position of tRNAGly (UCC) (*Leihne et al., 2011*; *Zdzalik et al., 2014*). Additionally, AtALKBH6 was considered as a potential eraser playing important roles in seed germination, seedling growth, and survival of *Arabidopsis* under abiotic stresses (*Huong, Ngoc & Kang, 2020*). Therefore, AtALKBH6/8-like proteins (BvALKBH5B, BvALKBH7B and BvALKBH9B) may participate in tRNA modification and DNA repair. The *Arabidopsis* AtALKBH1 protein was reported to be involved in organellar system of alkylation lesion repair (*Kawarada et al., 2017*; *Mielecki et al., 2012*). The members of Class V (AtALKBH1-like) were BvALKBH1B, BvALKBH2B and BvALKBH3B, which are possibly mostly associated with redox and tRNA modifications in cytoplasm and mitochondria.

The structure of a protein determines its primary function. Motif analysis of BvALKB proteins showed that almost all proteins had the motifs 1, 2, 4 and 8. The location of these motifs in each protein was consistent with that of the 2OG-Fe II-Oxy domain, suggesting that they are an important part of the conserved domain. Generally, most of the BvALKB

genes in one group had similar conserved motifs, which further supports the classification in the present study and the evolutionary relationships among the groups. The *A rabidopsis* AtALKBH9B and AtALKBH10B have been confirmed as $m^6A$ demethylases. Three proteins (BvALKBH6B, BvALKBH8B and BvALKBH10B) were homologous to AtALKBH9B/10B. These proteins shared motifs 3 and 6, which may be related to the demethylation function.

The expression profiles of demethylases in sugar beet leaves and roots under normal and salt stress conditions were analyzed. In leaves, all genes except *BvALKBH1B* were induced or inhibited by salt stress. In roots, five genes were up-regulated whereas five genes were down-regulated, and three genes were highly down-regulated. Apart from *BvALKBH5B* and *BvALKBH7B*, the other eight genes showed different expression trends in leaves and roots, suggesting tissue specificity of gene regulation. We gave further attention to the gene expression levels of three homologous proteins. *BvALKBH6B* and *BvALKBH8B* were down-regulated in leaves, but *BvALKBH10B* was significantly up-regulated, and the opposite trend was observed in roots. This work provides a molecular basis for further research on the mechanism of RNA methylation in regulating salt stress response in sugar beet.

## CONCLUSIONS

This study identified ten proteins of the ALKB family in sugar beet. Interestingly, the expression of BvALKBH10B was significantly up-regulated in leaves under salt stress, suggesting that it may play an important role in the response to salt stress mediated by RNA methylation modification. Finally, BvALKBH10B was cloned and sequenced. Our data will hopefully support future studies on RNA methylation in response to salt stress in sugar beet.

### Funding
This work was supported by the National Natural Science Foundation of China (No. 31571731). The funders had no role in study design, data collection and analysis, decision to publish, or preparation of the manuscript.

### Grant Disclosures
The following grant information was disclosed by the authors:
The National Natural Science Foundation of China: No. 31571731.

### Competing Interests
The authors declare there are no competing interests.

### Author Contributions
- Jie Cui conceived and designed the experiments, analyzed the data, authored or reviewed drafts of the paper, and approved the final draft.

- Junli Liu performed the experiments, analyzed the data, prepared figures and/or tables, and approved the final draft.
- Junliang Li performed the experiments, analyzed the data, authored or reviewed drafts of the paper, and approved the final draft.
- Dayou Cheng and Cuihong Dai conceived and designed the experiments, prepared figures and/or tables, and approved the final draft.

### DNA Deposition
The following information was supplied regarding the deposition of DNA sequences:
The gene sequence of BvALKBH10B is available at GenBank: MZ358117.

### Data Availability
The raw measurements are available in the Supplementary File.

### Supplemental Information
Supplemental information for this article can be found online at http://dx.doi.org/10.7717/peerj.12719#supplemental-information.

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
