# Peer review of "Genome-wide sequence identification and expression analysis of N6-methyladenosine demethylase in sugar beet (Beta vulgaris L.) under salt stress"

_PeerJ, doi:10.7717/peerj.12719_

## Round 0.1 · original submission · Major Revisions

Dear authors,

Based on the many concerns raised by the 3 expert reviewers in the field, concerns which I too share with the reviewers, I am returning your manuscript to you for MAJOR REVISIONS.

Please carefully consider the comments/concerns raised by all 3 reviewers when revising your manuscript for its resubmission.

Specifically, (1) carefully edit the text of all sections of your manuscript for its improvement; (2) perform appropriate statistical analysis on your experimental data; (3) reduce the current number of Figures from 9 down to 5 or less (via combining the individual Figures), and; (4) only make claims that can be supported by the experimental findings reported in your study.

A significant amount of additional work is required in order to improve the overall standard of your manuscript before its resubmission. Therefore, please take your time when addressing the many numerous issues relating to your originally submitted manuscript, before considering resubmission of a vastly improved study.

All the best,
Andrew Eamens

·

Basic reporting

no comment

Experimental design

no comment

Validity of the findings

no comment

Additional comments

The authors identified 12 sugar beet ALKB family proteins in sugar beet genome-wide database and analyzed its gene structures, chromosome location, physical and chemical properties of protein, motifs, subcellular localization and the phylogenetic tree construction etc, and quantitatively comparing the expression of BvALKB under normal conditions and salt stress. They claimed that these genes were in response to salt stress. However the manuscript suffered several problems. I cannot recommend for further processing.
Main concerns are:
1. The analysis of the physicochemical propertids of BvALKB proteins is oversimplified. The physicochemical propertid of BvALKB proteins should be validated by complementary experiments in vitro, eg. Prokaryotic expression of BvALKB proteins.
2. The authors claimed that BvALKB proteins were in response to salt stress. However, there is no other experimental evidence to prove that BvALKB proteins play a role in response to salt stresses except for quantitatively comparing the expression of BvALKB under normal conditions and salt stress.
3. There is no difference significance analysis was performed in quantitative PCR experiments.
4. The Fig. 1, 2 and 7 are not very clear.
5. The analysis of subcellular localization prediction of BvALKB proteins is oversimplified. eg, leaves of Nicotiana benthamiana could be used for transient transformation and subcellular lacalization analysis of BvALKB proteins.

In short, experimental data are missing to prove correlations between BvALKB proteins and salt stress. Thus, I suggest to accepting it for publication after major revisions.

Reviewer 2 ·

Basic reporting

These requirements are well.

Experimental design

Experimental design is reasonable

Validity of the findings

The findings are novel.

Additional comments

m6A, as a most abundant and highly conserved RNA modification, plays important roles in plant growth, development and response to abiotic stresses. m6A is regulated by Methytransferase, demethylase and readers proteins. Here, Cui et al genome widely identified the genes encoding m6A demethylases from sugar beet, they found out 12 genes and performed bioinformatic analysis and q-PCR experiment for some gene under salt stress. This study provides a theoretical basis for further screening of m6A demethylase in sugar beet, and also lays a foundation for studying the role of ALKB family proteins in growth, development and response to salinity stress. There are some concerns as below:
1. Please introduce the original and meaning about hush, pkhc,ryeg, et al; the gene need to be named according to homologous from Arabidopsis;
2. The 10 figures in MS need to be integrated less than 5 figures according to the result description;
3. The results of Qrt-PCR need provide significant analysis;
4. Some English writing need to be improved.

Reviewer 3 ·

Basic reporting

The study was aimed on the N6-methyladenine demethylase gene family in sugar beet. The sugar beet genome database was screened with 2OG-Fe II-oxy conserved domain of the ALKB demethylase family. A total of 12 proteins homologous to ALKB demethylases and the corresponding genes were identified. Several bioinformatic analyzes were performed to provide data on protein domain distribution, putative cellular localization, phylogenetic relationships within the BvALKB family, gene structure and the chromosomal localization. In my opinion bioinformatic analysis was performed properly and provides comprehensive characteristic of the ALKB demethylase family in sugar beet. Finally, the effect of salt stress on the transcript level was assessed using RT qPCR approach. It was revealed that one of the BvALKB transcripts, the “uxaj” seems to be strongly up-regulated under salinity treatment in sugar beet leaves.

Experimental design

1. The absence of statistical analysis of the gene expression results is a serious problem. Besides, the fold change in gene expression level between the control plants and the salt-treated plants the statistical significance must be determined and shown oh the graphs on Fig. 9. Especially as the Authors say several times that the expression of a given gene is significantly down-or up-regulated with respect to untreated control. The significance has to demonstrated by proper statistical analysis. It also has to be kept in mind that even if the difference is statistically significant the conclusions on the difference in gene expression should be cautious if the fold change is smaller than two.
2. What is actually shown on Fig. 9? Is it a mean from all three replicates or a representative result for a given experimental variant? Do the error bars show standard deviation or standard error.
3. It was said (l. 150-151) that the gene expression experiments were repeated at least three times, but with how many plants per replicate?
4. The fold change (not “data analysis”, l. 151) was calculated by deltaCt method. This method is applicable when the PCR efficiency of target gene amplification is comparable to the internal control (housekeeping gene) amplification. Otherwise, the Pfaffl’s method with the correction of difference in PCR efficiencies is recommended. Were the PCR efficiencies determined. If so, they should be shown in table 1.
5. Three different housekeeping genes were used to normalize the gene expression data. Which one was used to calculate the results shown on Fig. 9?
6. Please correct the l.163-165 by applying correct term to denote the discrete parts of the protein: The aminoacid chain has no “front”, “middle” and the “end”. It has N-terminus (or N-terminal domain), internal domain and C-terminal domain (C-terminus).

Validity of the findings

The study provides some insight into the largely neglected issue of the demethylase in sugar beet and their putative involvement in stress response. However, several topics have to be revised before publication.

Additional comments

1. Avoid unjustified speculation, which is not supported by the data such as : “njrf, eskg and uxaj are likely to […] play an important role in the response of sugar beet to salt stress” (l.261-262). This statement should be corroborated by functional study. What is shown here, is the correlation between stress treatment and gene expression.
2. The English language should be improved, throughout. Several terms are used inappropriately and incorrectly, such as “coping with different environments” (l.41), “The reversibility of m6A […] was confirmed in the paper” (l.55) “structures were analysis” (l.136), “transcriptional protein”?? (l. 261). Moreover please correct: “identified” instead of “screened” (l.158), “sugar” instead of “suagr” (l. 184),

---

## Round 0.2 · Minor Revisions

Many numerous English language issues remain in this revised version of your original submission.

Please address these issues in a substantially revised manuscript version.

---

## Round 0.3 · Minor Revisions

As requested as part of the previous round of revision, English language issues persist in the revised version of our manuscript.

These issues must be addressed prior to this manuscript proceeding any further along the line of publication acceptance.

I do not have any remaining concerns with the science being reported on in your study, it is simply now a matter of improving the standard of the English language.

Please either have your manuscript reviewed by a fluent English-speaking colleague or employ an English language editing service to make the textual changes required.

Again, I thank you for the efforts made to date which have significantly improved the standard of your study, however, this remaining concern must still be addressed.

Kind regards,
Andy.

---

## Round 0.4 · Minor Revisions

Dear authors,

I am again going to have to class this version of your manuscript as requiring additional MINOR corrections. Please see outstanding issues listed below;

1- A small number of text related issues remain - please see that these issues are addressed via use of the attached annotated PDF.

2 - Figure 1A and 1B only report on 9 gene family members, whereas throughout the manuscript all ten family members are reported on - why the difference in number? Also, Figure 1C goes back to reporting on all 10 family members - therefore, again, why does Figs 1A and 1B only include the 9?

3 - The Material and Methods requires a section on the statistical analyses that you used to determine significant differences in gene expression presented in Figure 5. Furthermore, the gene expression changes presented in Figure 5 cover an extensive range of differences from gene to gene analysed. Therefore, you must also show degrees of significance, not just yes or no for statistical significant differences (i.e., denote on the graph columns *, **, *** and state the degree of significance for each in the Figure 5 legend).

Cheers,
Andrew.

---

## Round 0.5 · accepted · Accept

Dear authors,
Thank you kindly for your repeated efforts throughout this editorial process: your willingness to continue to modify your manuscript as requested has been highly appreciated.

The repeated revisions made to your original submission have also improved the level of your study to publication acceptance.

Thank you again for your efforts and interactions in taking your study through the editorial process.

Kind regards,
Andrew